# Pretransplant BK Virus-Specific T-Cell-Mediated Immunity and Serotype Specific Antibodies May Have Utility in Identifying Patients at Risk of BK Virus-Associated Haemorrhagic Cystitis after Allogeneic HSCT

**DOI:** 10.3390/vaccines9111226

**Published:** 2021-10-22

**Authors:** Markéta Šťastná-Marková, Eva Hamšíková, Petr Hainz, Petr Hubáček, Marie Kroutilová, Jitka Kryštofová, Viera Ludvíková, Jan Musil, Pavla Pecherková, Martina Saláková, Vojtěch Šroller, Jan Vydra, Šárka Němečková

**Affiliations:** 1Institute of Haematology and Blood Transfusion, 128 20 Prague, Czech Republic; marketa.markova@uhkt.cz (M.Š.-M.); eva.hamsikova@gmail.com (E.H.); petr.hainz@uhkt.cz (P.H.); marie.kroutilova@uhkt.cz (M.K.); jitka.krystofova@uhkt.cz (J.K.); viera.ludvikova@uhkt.cz (V.L.); jan.musil@uhkt.cz (J.M.); pavla.pecherkova@uhkt.cz (P.P.); Martina.Salakova@uhkt.cz (M.S.); vojtech.sroller@uhkt.cz (V.Š.); jan.vydra@uhkt.cz (J.V.); 2Laboratory of Virology, Motol Hospital, 150 06 Prague, Czech Republic; petr.hubacek@lfmotol.cuni.cz; 3Department of Genetics and Microbiology, Faculty of Science, Charles University, 120 00 Prague, Czech Republic; 4BIOCEV, Biotechnology and Biomedicine Center of the Academy of Sciences, Charles University, 252 50 Vestec, Czech Republic

**Keywords:** human BK polyomavirus 1 and 4 (BKPyV 1 and 4), haemorrhagic cystitis, anti/BKPyV IgG, T-cell response, VP1, LTag, immunocompromised patients, hemopoietic stem cell transplantation

## Abstract

BK polyomavirus (BKPyV) persists lifelong in renal and urothelial cells with asymptomatic urinary shedding in healthy individuals. In some immunocompromised persons after transplantation of hematopoietic stem cells (HSCT), the BKPyV high-rate replication is associated with haemorrhagic cystitis (HC). We tested whether the status of BKPyV immunity prior to HSCT could provide evidence for the BKPyV tendency to reactivate. We have shown that measurement of pretransplant anti-BKPyV 1 and 4 IgG levels can be used to evaluate the HC risk. Patients with anti-BKPyV IgG in the range of the 1st–2nd quartile of positive values and with positive clinical risk markers have a significantly increased HC risk, in comparison to the reference group of patients with “non-reactive” anti-BKPyV IgG levels and with low clinical risk (LCR) (*p* = 0.0009). The predictive value of pretransplant BKPyV-specific IgG was confirmed by determination of genotypes of the shed virus. A positive predictive value was also found for pretransplant T-cell immunity to the BKPyV antigen VP1 because the magnitude of IFN-γ T-cell response inversely correlated with posttransplant DNAuria and with HC. Our novel data suggest that specific T-cells control BKPyV latency before HSCT, and in this way may influence BKPyV reactivation after HSCT. Our study has shown that prediction using a combination of clinical and immunological pretransplant risk factors can help early identification of HSCT recipients at high risk of BKPyV disease.

## 1. Introduction

Human BK polyomavirus (BKPyV) is a highly prevalent virus that establishes latency in the urinary tract after usually asymptomatic primary infection early in life. BKPyV isolates are classified into four major subtypes and many subgroups based on the VP1-DNA sequence [1]. Subtype I is prevalent worldwide (80%), followed by subtype IV (15%), which is common in Asia and parts of Europe. Subtypes II and III occur less frequently [2]. Seroepidemiology studies have shown that antibody responses specific for subtype I, including all subgroup variants, can be found in up to 98% of adults [3,4]. Seroreactivity to subtypes II, III and IV in healthy donors was 86%, 77% and 80%, respectively. Detection of the high seroprevalence of rare serotypes II and III could be ascribed to cross-reactivity with serotype IV. Cross-reactivity was also observed among subgroups of serotype I [5]. A seroepidemiologic study of BKPyV in the Czech Republic has shown that the seropositivity rate of 80% that was found for young adults decreased with age to 56% [6], indicating that virus latency can be associated with the waning of specific humoral immunity. BKPyV infection is not associated with any pathology in immunocompetent individuals, but in immunocompromised patients after transplantation of a kidney or haematopoietic stem cells (HSCT), virus reactivation/reinfection can cause serious clinical illness. BKPyV reactivation after allogeneic HSCT can cause haemorrhagic cystitis (HC) or nephropathy (for an overview, see [7]). BKPyV infection is responsible for haemorrhagic cystitis (HC) in up to 17% of recipients of allogeneic HSCT within the first year after treatment [8]. BK virus DNA can be detected in the urine of up to 80% of patients after allogeneic HSCT [9,10]. However, patients who develop symptoms of HC excrete significantly higher levels of BKPyV DNA in urine [11]. Reported risk factors of HC are associated with immunosuppression and damage to the urinary tract epithelium caused by chemotherapy. They include cord blood HSCT, myeloablative conditioning (MAC) [12,13,14,15,16], chemotherapy with cyclophosphamide, anti-thymocyte globulin (ATG) administration and graft versus host disease (GvHD) therapy. As no effective antiviral treatment of BKPyV infection in HSCT patients is currently available, BKPyV-specific adoptive T-cell therapies were developed [17,18].

The value of pretransplant BKPyV-specific antibodies in transplanted patients for protection from viral disease has long been considered insignificant. Recently, subtype-specific antibodies against BKPyV VP1 have been reported to have a subtype-specific virus-neutralising activity in kidney transplant patients [19]. The predictive value of pretransplant anti-BKPyV IgG in allo-HSCT recipients is less evident. Positive pretransplant anti-BKPyV IgG has been regarded as a significant predictive value for viruria [20,21,22]. However, it has been observed that very high levels of anti-BKPyV IgG (titer > 1:40,960) prior to allo-HSCT can be associated with lower peak viremia in the first 100 days after HSCT, whereas anti-BKPyV IgG titers below 1:40,960 were associated with higher grade viremia [23].

Cellular immune responses against BKPyV antigens in HSCT patients were examined in a small number of studies [24,25], usually after transplantation in context with BKPyV reactivation and HC. We hypothesised that the status of BKPyV immunity prior to HSCT could provide evidence for the BKPyV tendency to reactivate, and that examining the level of subtype-specific antibodies and T-cell response in individual patients could help predict the risk of BKPyV reactivation and HC. To evaluate the risk of HC in relation to clinical factors known before transplantation, we analysed a large cohort of HSCT recipients. The predictive value of BKPyV specific immunity in combination with clinical risk factors of HC was then confirmed in a smaller cohort of transplanted patients (PBIHC).

## 2. Materials and Methods

### 2.1. Patients, Donors and Sample Collection in PBIHC Study (Pretransplant BKPyV—Specific Immune Response for HC Risk Assessment)

HSCT recipients (*n* = 149) and their HSC donors (*n* = 120) were invited to participate in the PBIHC study at the Institute of Haematology and Blood Transfusion (IHBT) in the period of 2017–2020. Pretransplant serum/plasma samples of every patient or donor were isolated and stored in aliquotes at −20 °C. All transplanted patients obtained peripheral blood progenitor cells as a graft. Urine samples were collected from most of the patients during the first month after HSCT; afterwards, sampling was based on clinical indication. During the PBIHC study, 37 patients donated an additional blood sample one week before transplantation for detection of BKPyV-specific T-cell response.

### 2.2. Haemorrhagic Cystitis Diagnosis

Twenty-two of the 149 patients developed HC grade 2 or higher based on clinical symptoms. Grade 2 HC was diagnosed in three, grade 3 in sixteen and grade 4 in three of the patients. An additional 12 of the 149 patients had HC grade 1. Grade 1 was defined as microscopic haematuria, grade 2 as macroscopic haematuria, grade 3 as macroscopic haematuria with small clots and grade 4 as massive haematuria requiring instrumentation for clot evacuation and/or causing urinary obstruction [8]. Ninety percent of patients were examined for BKPyV DNAuria at least once after HSCT. Diagnosis of BKPyV-associated HC was in most cases confirmed by repeated measurements of the BKPyV viral load. Viral DNA was determined by a real-time polymerase chain reaction (qPCR) in the Laboratory of Virology of the Motol Hospital, Prague. The viral load was expressed as copies per mL/urine [26,27]. Quantitative measurement of DNA viremia was performed in just six patients with suspicion of nephropathy.

### 2.3. Algorithm for Assessment of HC Risk upon Patient Demographic and Clinical Pretransplant Characteristics

To identify potential risk factors associated with HC and to determine the predictive value of data concerning age, gender, diagnosis, donor type, conditioning regimen intensity, GVHD, ATG or posttransplant cyclophosphamide (PTCP), we retrospectively analysed the clinical data of all 524 patients who underwent allogeneic HSCT at the IHBT in years 2014–2020 (Table 1).

The HC incidence in this cohort was 13% (Table 1). The values in the individual years varied randomly with a standard deviation of ±2.92. In univariate analysis, the following variables were associated with increased risk of HC: age, male gender, MA conditioning, MUD transplant with use of ATG. A decreased risk of HC was associated with the female gender, reduced intensity (RI) conditioning and MRD transplant. No risk was associated with acute GVHD and prevailing underlying diagnoses, such as acute myeloid leukaemia (AML), myelodysplastic syndrome (MDS) and lymphatic leukaemias. Increased risk of HC was found in patients with chronic myeloid leukaemia (CML) (6 of 16) and acute promyelocytic leukaemia (2 of 3), which were very rarely treated with HSCT. Based on these pretransplant clinical data, patients were divided into a low clinical risk group (LCR) comprising all female recipients, recipients with RIC and recipients of graft from 10/10 MRD and a group with high clinical risk of HC (HCR), comprising the remaining patients. The hazard ratio of HC between these two groups differed significantly (Table 1, HR = 2.955. 95%CI = 1.890–4.607. *p* < 0.0001). HC risk prediction from these clinical data did not enable the risk associated with BK virus infection and its reactivation to be determined, as serological data were not available. To assess the risk of HC based upon clinical characteristics and BKPyV, this method was then used for patients (*n* = 149) in the PBIHC study (Table 2).

**Table 2 vaccines-09-01226-t002:** Risk of HC in the PBIHC study cohort of HSCT recipients 2017–2020 (*n* = 149) including clinical features and pretransplant BKPyV-specific antibodies.

PBIHC Study				Risk of HC		
				Univariate Associations	Multivariate Associations	
Parameter	HC	Non HC	HC Rate [%]	^a^*p* Value	^b^ χ^2^ Test	Logistic Regression
HSCT recipients (*n* = 149)	22	127	14.8			
Maximum post HSCT BKPyV viruria						
ref < 107 copies/mL	0	90	0			OR = 9.149 95%IC = 2.717–62.75
>10^7^ copies/mL	22	37	37.2.0	*p* < 0.0001 HR = 4.233 95%IC = 4.233–5.709		
Pretransplant conditioning						
MAC	19	92	17.1	*p* = 0.1963		n.s.
RIC	3	35	7.9			
Gender						
Male	18	74	19.5	*p* = 0.055		n.s.
Female	4	53	7.0			
Clinical risk group						
HCR (male MAC + MUD + MMRD)	15	43	25.8	*p* = 0.0038 HR = 3.362 95%IC = 1.500–7.613		
LCR (all RIC + all MRD female MAC + MUD + MMRD)	7	84	7.6			
Pretransplant anti BKPyV1,4 IgG ref = NR^+^ level						
^1^ “NR^+^” levels anti-BKPyV IgG	5	50	9.0		*p* = 0.2411	
^2^ “R” levels anti-BKPyV IgG	14	57	19.7	*p* = 0.1878		
^3^ “R_75_”anti-BKPyV IgG	3	20	13.0	*p* = 0.4878		
Combined risk HC ref = LCR “NR^+^” BKPyV IgG						
LCR ^1^ “NR^+^” anti-BKPyV IgG	1	34	2.8		*p* = 0.1603	
LCR ^2^ “R” anti-BKPyV IgG	4	36	10.0	*p* = 0.3636		
LCR ^3^ “R_75_” anti-BKPyV IgG	2	14	12.5	*p* = 0.2286		
HCR ^1^ “NR^+^” anti-BKPyV IgG	4	16	20.0	*p* = 0.0532		
HCR ^2^ “R” anti-BKPyV IgG	11	21	34.7	*p* = 0.0009 HR = 0.4165 95%CI = 0.2832–0.6466		
HCR ^3^ “R_75_” anti-BKPyV IgG	0	6	0.0	*p* > 0.999		

^a^ Comparison with reference group using Fisher’s exact test. ^b^ Association of HC with risk factors was analysed by the χ^2^ test. ^1^ Samples included in AZ rectangle in Figure 1A. ^2^ Samples included in AY. BY. CY. BX. BZ rectangle in Figure 1A. ^3^ Samples included in AX. CX. CZ rectangle in Figure 1A. Abbreviations: LCR—low clinical risk (all RIC + all MRD female MAC + MUD + MMRD). HCR—high clinical risk (male MAC + MUD + MMRD). HR—hazard ratio. RIC—reduced intensity conditioning regimen. MAC—myeloablative conditioning regimen. MUD—matched unrelated donor. MMRD—mismatched related donor. AML—acute myeloid leukaemia. MDS—myelodysplastic syndrome. ALL LBL—acute lymphoblastic leukaemia/lymphoblastic lymphoma. nHL—B-non Hodgkin lymphoma. CLL—chronic lymphocytic leukaemia. CML—chronic myeloid leukaemia. AA—aplastic anaemia.

**Figure 1 vaccines-09-01226-f001:**
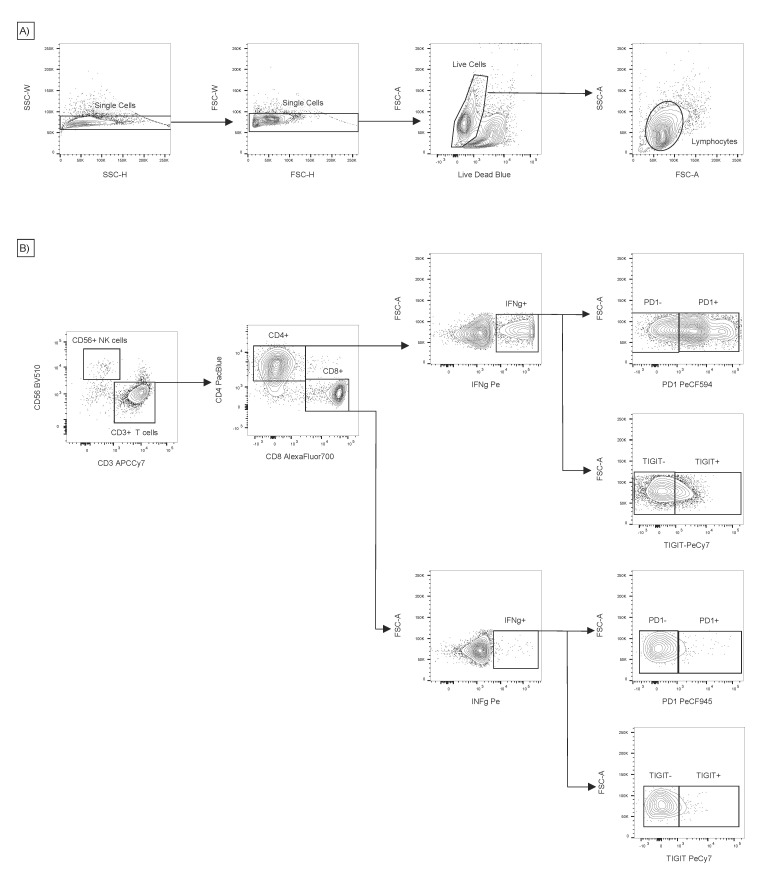
Gating strategy for IC-FACS analysis. (**A**) Doublets, dead cells and debris were removed from the samples and lymphocytes were gated based on SSC and FSC. (**B**) NK cells and T-cells were separated using CD56 and CD3. The expression of IFNγ by CD4 and CD8 T-cells was subsequently evaluated. The presence of PD1 and TIGIT exhaustion markers was detected on the on CD4 + IFNγ+ and CD8 + IFNg + cells.

### 2.4. BK and JC Virus-Like Particles

The VP1 coding sequences of BKPyV 1, 2, 4 and JCPyV, all codon-modified for expression in human cell lines, were synthesised by the GeneArt Gene Synthesis service (Thermo Fischer Scientific, Carlsbad, CA, USA). The coding sequences for VP1 proteins of BKPyV1-D, subtype I/a (GenBank accession number: JF894228), BKPyV2-GBR-12 (GenBank accession number: AB263920), BKPyV4-A-66H, subtype IV/c-2 (GenBank accession number: AB369093), all with a length of 362 amino acids and JCPyV with a length of 354 amino acids (strain Mad1, GenBank accession number: J02226.1) were inserted into the pFastBacTM Dual plasmid (Life Technologies) downstream of the polyhedrin promoter. The integrity of the inserted gene was confirmed by sequencing. Recombinant baculoviruses that expressed the VP1 genes were prepared according to the Bac-to-Bac Baculovirus Expression System manual (Thermo Fischer Scientific). Recombinant baculoviruses were plaque purified. The production of virus-like particles (VLP) of human polyomaviruses in SF-9 cells has been described previously [28]. The integrity of the VLPs of individual polyomaviruses was verified by electron microscopy.

### 2.5. Measurement of BK and JC Virus Specific Antibodies

The presence of antibodies to BKPyV1, 2, 4 and JCPyV was tested using an in-house enzyme-linked immunosorbent assay (ELISA) based on VP1 virus-like particles (VLP). Briefly, wells of microtiter plates (Polysorp; NUNC, Thermo Fischer Scientific were coated with purified VLPs in PBS (100 ng/well) at 37 °C for 2 h and at 4 °C overnight. All subsequent incubations were performed at 37 °C for 1 h. The wells were repeatedly washed using the 1575 automatic microplate washer (Bio-Rad Laboratories, Hercules, CA, USA) with buffer A (PBS, 0.21 mol/L NaCl and 0.1% Triton X-100) to remove unbound reagents. Nonspecific binding sites were blocked by incubation with 1% BSA in PBS, and the wells were subsequently incubated in duplicate with human sera diluted 1:100 in buffer A with 1% BSA. Following incubation, antibodies bound were detected with Peroxidase-AffiniPure Donkey Anti-Human IgG (H + L) (Jackson ImmunoResearch Europe Ltd., Ely, UK), and the reaction was visualised by adding 100 μL of o-phenylenediamine containing a substrate solution. The colour reaction was stopped by 100 μL of 2 mol/L H_2_SO_4_, and optical densities (ODs) at 492/630 nm were read using the Infinite 200 plate reader (Tecan Trading AG, Männedorf, Switzerland). Background reactivity was determined in wells without antigen. Their absorbances were subtracted from the corresponding values obtained in the presence of the antigen. Positive, negative and cut-off controls for the corresponding antigen were tested on each plate. All ELISA results were represented as a ratio of the absorbance obtained with the tested sample and cut-off control (OD index-OI), which expresses the strength of the antibody response. Samples with an OI of ≤1 were considered nonreactive. All samples within 10% above the cut-off (CO) value, as well as about one-quarter of all serum samples, were retested to confirm the results. Only concordant results 10% above the CO value were classified as reactive. The test was repeated for the third time in the case of discordant results of samples initially reactive/negative. The final result was concordant in two of the three repeated tests. About 500 sera from healthy blood donors were tested in ELISA with BKPyV1, 2, 4 and JCPyV derived VLPs, to distinguish between negative and reactive samples. The CO value was calculated for each antigen as described earlier [6,29].

For the purposes of analysing pretransplant anti-BKPyV IgG levels against BKPyV1 and BKPyV4 of HSCT recipients, the samples were classified into three groups denoted “non reactive plus” (NR^+^), “reactive” (R) and “75th percentile of reactive” (R_75_) using thresholds T1 and T2. Threshold T1 was determined as the arithmetic mean of the seronegative OI values + 3 standard deviations. The value of T1 was 1.35 and 1.25 for BKPyV1 and 4, respectively. The OI values lower than T1 comprising non-reactive and “grey zone” samples were assigned as “NR^+^”, and are plotted in rectangle AZ of Figure 2C. The OI values lying between thresholds T1 and T2 (plotted in rectangles AY, BY, CY, BX, BZ in Figure 2C) were denoted as reactive “R”. Threshold T2 was determined as the value of the 75th percentile of positive samples (OI > 1) and was 2.69 and 2.73 for BKPyV1 and BKPyV4, respectively. The values of patients with “R_75_” levels of pretransplant antibodies against BKPyV1 or BKPyV4 exceeding the T2 threshold were plotted in rectangles AX, CX, CZ in Figure 2C.

### 2.6. Detection of BKPyV-Specific T-cells

PBMCs were isolated from the blood of patients and HSC donors on the Ficoll-Paque Plus gradient (GE Healthcare, Uppsala, Sweden). The fresh cells were washed with PBS containing 2% human serum albumin. They were subsequently stimulated for 1 h with a mixture of 15 amino acid long overlapping peptides (PepMix^TM^) covering BKPyV VP1 and BKPyV LTAG (JPT Peptide Technologies GmbH, Berlin, Germany). The PepMixes were used at concentrations of 0.1 µg per 15 million PBMC. The stimulated cells were resuspended in 15 mL of culture medium (CTL) composed of 50% RPMI 1640 with HEPES and glutamine, 45% Click’s medium (both from FUJIFILM Irvine Scientific, Santa Anna, CA, USA), 5% human AB-serum (Capricorn Scientific, Ebsdorfergrund, Germany), 1% penicillin-streptomycin-glutamine (GIBCO, Dublin, Ireland) and supplemented with IL-4 and IL-7 (CellGenics, Freiburg, Germany) at a concentration of 16.6 ng and 10 ng/mL, respectively. They were then cultured in 45 cm^2^ culture flasks (Corning, Corning, NY, USA) under 5% CO_2_ at 37 °C. On day 5 of culture, 10 mL of exhausted medium were replaced with the fresh CTL containing 25 ng/mL of IL-4 and 15 ng/mL of IL-7. The culture was harvested on the 12th day, and the frequency of the BKPyV-specific T-cells among expanded T-cells was measured by ELISPOT-IFNγ or by flow cytometry with intracellular cytokine staining (IC FACS).

### 2.7. ELISPOT-IFNγ

Expanded T-cells were resuspended in CTL supplemented with 0.5 µg/mL costimulatory molecules (anti-CD28 and anti-CD49d, BD Bioscience) to a concentration of 3.5 × 10^5^ live cells/mL. From this cell suspension, 0.2 mL/well were seeded in triplicates on ELISPOT plates (MIPN 4550. Sigma-Aldrich, Prague, Czech Republic) precoated with an anti-interferon γ capture antibody (anti-IFN γ, clone 1-D1K, 0.75 µg/well (MabTech, Nacka Strand, Sweden)). Restimulation was performed by individual PepMixes (1 µg/mL) in triplicates under 5% CO_2_ at 37 °C for 20 h. Stimulation with purified anti-human CD3 mAb CD3-2 (anti-CD3) 10 µg/mL (MabTech) was used as the positive control. Cells incubated without stimulants served as the negative control. At the end of the stimulation, the cells were washed out. IFNγ spots were stained with a 1 µg/mL biotinylated anti-IFNγ detection antibody (clone 7B-6-1-Bi. MabTech), streptavidin-horseradish peroxidase (HRP) conjugate 10 µg/mL (Streptavidin HRP, MabTech) and AEC substrate (BD Bioscience) according to the manufacturer’s protocols. The spot-forming cells (SFC) were counted using the Immunospot analyser (Cellular Technology Limited CTL, Cleveland, OH, USA). The mean SFC numbers were normalised for 1.25 × 10^5^ cells. The SFC values in negative controls were subtracted from those determined in stimulated cultures.

### 2.8. Intracellular Cytokine Detection by Flow Cytometry (IC FACS)

Lymphocytes 5 × 10^6^/mL were stimulated with 1 μg/mL PepMix in a CTL medium for 2 h and then for an additional 12 h in the presence of 1:1000 diluted Golgi plug (BD Biosciences, Franklin Lakes, NJ, USA). After incubation, cells were harvested with PBS and stained with the LIVE/DEAD Blue Dead Cell Staining Kit (Thermofisher) and antibodies to CD45RA-BB515, CD27-BV650, CD45RO-BV786, PD1-PECF594, CCR7-BV605 (BD Horizon, BD Biosciences), CD56-BV510, TIGIT (VSTM3)-PE/Cy7 (Biolegend, San Diego, CA, USA), CD8-AlexaFluor700 (Exbio, Prague, Czech Republic) and CD57—APC (BD Pharmingen). The cells were then washed with PBS, fixed using IC fixation and permeabilisation Buffer (eBioscience, San Diego, CA, USA) for 20 min and stained intracellularly with antibodies against IFN-gamma-PE, CD3-APCCy7 (Biolegend, San Diego, CA, USA) and CD4-PacificBlue (Exbio) in a permeabilisation buffer. The cells were washed and resuspended in a FACS buffer (FB-PBS containing 0.09% sodium azide, 1% BSA). The cells were measured using the BD LSR Fortessa 5 L flow cytometer (BD Biosciences). The obtained data were analysed by the FlowJo 10.5 software (TreeStar, Ashland, OR, USA). The gating strategy is shown in Figure 1.

### 2.9. BKPyV Genotyping

Viral subtype identification was performed as described by Morel V et al. [30]. In brief, BKPyV genotyping was based on a 100 bp segment from nucleotide region 1977 to 2076 within VP1. Viral DNA was first amplified using primers BKV-S1920 5′-GGTYATTGGAATAACTAGYATGC-3′ and BKV-A2159 5′-TCCAARTAGGCCTTATGRTCAG-3′. Sanger sequencing of the purified amplicons was performed using the Applied Biosystems BigDye Terminator v3.1 kit according to the manufacturer’s instructions. The same primers were used individually for the generation of forward and reverse sequences, which were read with the use of an Applied Biosystems 3500 Genetic Analyser. The nucleotide sequences obtained were then manually reviewed and used for subtyping using a described algorithm. The DNA of a plasmid pBKV (34-2), subtype BKPyVI/a, kindly provided by J. Forstová, Charles University, Faculty of Science, Prague, served as a positive control.

### 2.10. Statistics

Differences in categorical and continuous variables were assessed with the Fisher exact and Mann–Whitney or Wilcoxon matched-pairs signed rank test, respectively. Association of continuous variables was evaluated by the Spearman correlation analysis. Multivariate analysis was performed using the logistic regression model. Cumulative incidence curves were compared using the LogRank (Mantel–Cox) test. The tests were conducted at a level of significance higher than 0.05. All calculations and data plots were performed using the GraphPad Prism software, version 9.1.0 for Windows, GraphPad Software, San Diego, CA, USA.

## 3. Results

### 3.1. Polyomavirus-Specific Antibodies before HSCT in Recipients and Donors

To find an independent risk marker of the BK virus infection and HC, antibodies against BKPyV1,2,4 and JCPyV were measured in patients and donors before HSCT (PBIHC study). Specific seropositivity against BKPyV1, BKPyV2, BKPyV4 and JCPyV was found in 65%, 46%, 41% and 50% of patients and in 65%, 49%, 44% and 47% of HSC donors, respectively. The differences between the medians of the OI (see Material and Methods) were significant between BKPyV1 vs. BKPyV2 or BKPyV4 (*p* < 0.0001) for both the recipient and donor (Figure 2A). Negativity against any BKPyV found in patients (26%) and donors (28%) was not significantly different (Fisher’s test *p* = 0.6797). The seroprevalence found in the PBIHC study group (73%) was not significantly different from the seroprevalence found previously in the general local population [6]. However, in some seronegative patients, detection of IFNγ T-cell response before HSCT or detection of BKPyV DNA in urine shortly after HSCT gave evidence that those patients had experienced BKPyV infection (not shown). Their seronegativity could most probably be ascribed to a waning specific humoral response, though actual infection cannot be ruled out either. We analysed whether anti-BKPyV IgG levels were age dependent. The difference between younger (aged 19–47) (*n* = 76), with median 1.536 and 0.7750, and older (aged 48–69) (*n* = 73) patients, with median 1.410 and 0.800, for BKPyV1 and BKPyV4, respectively, was not statistically significant (*p* = 0.2942 and *p* = 0.7830 for BKPyV1 and BKPyV4, respectively) as tested by the Mann–Whitney test and by correlation analysis (Figure 2B).

We tested whether the level of pretransplant polyomavirus-specific antibodies was associated with DNAuria during the first year after HSCT. The maximum level of virus DNA in urine and OI of virus-specific antibodies correlated significantly (Figure 2B) for BKPyV4 (Spearman correlation Rs = 0.21; *p* = 0.008) and BKPyV2 (Rs = 0.19; *p* = 0.015). The DNAuria level did not correlate with levels of anti-BKPyV1, JCPyV IgG of recipients and with donor´s antibodies of any specificity. The values of antibodies specific for any BKPyV type mutually correlated significantly (*p* < 0.00001). A very high correlation between BKPyV2 and BKPyV4 specific antibodies was found in recipient (Rs 0.85) and donor (Rs = 0.9) sera. This was most probably caused by high amino acid VP1 sequence homology, and for this reason, the role of anti-BKPyV2-IgG was not further studied. The possible protective or causative role of the donor immunity against BKPyV reactivation or HC induction was determined in a subgroup of recipients who were seropositive for any BKPyV1, 2, 4 (*n* = 90). Viruria of >10^7^ copies/mL was detected in recipients of grafts from donors with an anti-BKPyV IgG OI of <1 and >1 in 40% and 46% of cases, respectively. Haemorrhagic cystitis occurrence was 20% if the donors were BKPyV IgG and either seroreactive or seronegative. This shows that donor’s anti-BKPyV IgG does not correlate with the incidence of viruria in HSCT recipients, and suggests that donor antigen-specific T-cell immunity is not protective or is not associated with HC pathology. Our findings do not rule out the possibility that HC pathology is associated with the reconstitution of the innate immune response to BKPyV replication. As JCPyV-specific T-cells are cross-reactive towards BKPy-VP1 protein and vice versa, we looked for a possible protective effect of JCPyV-specific immunity of recipients or donors. Cross-protection was ruled out by correlation analysis in Figure 2B. To confirm the predictive value of pretransplant BKPyV-IgG, we determined BKPyV genotypes in samples of those HSCT recipients with viruria exceeding 10^9^ genome equivalents/mL (*n* = 34). Sixty-two percent (61.8% (21)) of patients were positive for BKPyV I/b-2; 11.8% (4) for BKPyV I/b-1; 2.9% (1) for BKPyV III; and 23.5% (8) for BKPyV IV/c-2. The differences between serotype specific IgG levels were significantly different for patients with BKPyV I/b-2 and IV/c-2 (Figure 2E). Higher pretransplant serum antibody levels (BKPyV1 or BKPyV4) were directed against that virus genotype which was detected after the HSCT in patients’ urine. The association was statistically significant (*p* = 0.0253 for subgroup I/b-2 and *p* = 0.0347 for subgroup IV/c-2). Our results suggest that the patient is more likely to shed the same virus type against which highest BKPyV IgG levels were found.

### 3.2. “R” Levels of Pretransplant BKPyV-Specific IgG of Patients at High Clinical Risk (HCR) Are Associated with Increased Risk of HC

Further, we asked whether the levels of pretransplant anti-BKPyV-specific IgG of patients affected the incidence of HC. The levels of pretransplant anti-BKPyV1,4-IgG of individual patients in the PBIHC study are shown in Figure 2D. The large symbols stand for patients with HC. The overall results (Table 2) show HC to be less common (9.0%) in patients with “NR^+^” levels of pretransplant antibodies. This cluster includes seronegative patients with an OI of <1 as well as those with antibody levels below the T1 threshold level. The group of patients with the highest frequency of HC (19.7%) had “R” antibody levels. The patients with “R_75_” levels of pretransplant anti-BKPyV1,4-IgG antibodies were not at an increased risk of HC (13.0%). Hence, the incidence of HC did not significantly differ between groups stratified just by antibodies (Table 2). To reveal the risk of HC, the method for HC risk assessment based upon clinical characteristics verified on a large patient cohort (Table 1) (*n* = 524) was applied to patients from a recent PBIHC clinical study (Table 2) (*n* = 149), stratified according to their anti-BKPyV-IgG levels. This approach improved HC risk prediction, with the HCR “R” anti-BKPyV IgG cohort being at a significantly increased HC risk than the reference LCR “NR^+^” anti-BKPyV IgG cohort (Table 2, HR = 0.4165; CI95% = 0.2832–0.6466 (*p* = 0.0009). The advantage of this attitude lies in the fact that HC prediction is limited to patients infected with BKPyV.

Analysis of the cumulative incidence of HC in HSCT recipients during the first year after transplantation stratified according to anti-BKPyV-IgG levels and clinical risk factors revealed a significantly higher HC incidence in the HCR group with “R” levels of anti-BKPyV IgG in comparison with the LCR group (Figure 1E) (χ^2^ = 9.078, *p* = 0.0026), and non-significantly higher HC incidence than HCR “NR^+^” and “R_75_” groups.

### 3.3. Pretransplant Non-Specific and BKPyV-Specific T-Cell Response Is Low in Patients with High BKPyV Viruria after HSCT

We wanted to know whether the state of BKPyV-specific adaptive cellular immunity prior to HSCT influenced reactivation of BKPyV infection. Therefore, in addition to humoral response detection, the T-cell response against BKPyV VP1 and large T antigen was measured in patients recruited in the PBIHC study consecutively between V/2019 and II/2020. The evaluation was performed only in patients (*n* = 34) with at least one positive anamnestic marker of BKPyV infection (i.e., in those positive for pretransplant anti-BKPyV IgG, early DNAuria or positive pretransplant T-cell response). Before the start of conditioning, the patients were examined for T-cell response against VP1 and LTag (Figure 3A). T-cell response was compared between groups of patients stratified according to the posttransplant level of BKPyV DNAuria. The analysis revealed that the T-cells of patients with posttransplant DNAuria of <10^7^ BKPyV DNA copies/mL responded significantly better to stimulation with anti-CD3 (*p* = 0.0111) and with VP1 peptide pool (*p* = 0.0362) than the T-cells of patients with posttransplant DNAuria of >10^7^ BKPyV-DNA copies/mL. The absence of DNAuria in the first group was associated with the absence of HC (0/19), whereas the occurrence of DNAuria was associated with HC (10/15). The difference in the HC rate was statistically significant (*p* < 0.0001, Fisher’s exact test). One possible explanation of our results could be that the control of BKPyV latency by antigen-specific T-cells before HSCT would be one of the factors that influence BKPyV reactivation after HSCT.

The results of ELISPOT-IFNγ were confirmed by IC FACS-IFNγ in samples with sufficient numbers of in vitro expanded T-cells (*n* = 32). The analysis revealed that BKPyV VP1 or LTag antigens were recognised by CD4^+^ and CD8^+^ T-cells with a significantly prevailing CD4^+^ response (Figure 2B). Inhibitory molecules PD1 and TIGIT were determined on IFNγ-producing T-cells responding to stimulation with BKPyV antigens or the anti-CD3 antibody (Figure 3C). The expression of TIGIT (marker of anergy) on BKPyV-specific IFNγ^+^ CD4^+^ T-cells restimulated with anti-CD3 or VP1 was significantly decreased if they were from patients with low posttransplant DNAuria (<10^7^ BKPyV-DNA copies/mL), in comparison with the cells of patients with high posttransplant DNAuria of >10^7^ BKPyV-DNA copies/mL. This could mean that a less functional T-cell response before HSCT leads to higher viruria after HSCT. Quantification of PD1 and TIGIT was performed by FACS using the gating strategy depicted in Figure 1. Phenotype analysis of T-cells revealed that responding IFNγ^+^ cells were mostly effector memory T-cells of phenotype CD45RA-RO^+^ CCR7^−^ in 80% and 45% of CD4^+^ and CD8^+^ T-cells, respectively. Between 15% and 20% were central memory T-cells CD45RA-RO^+^ CD27^+^ CCR7^+^ (not shown).

## 4. Discussion

We analysed pretransplant clinical conditions and the BKPyV-specific adaptive immunity of recipients of allo-HSCT, which could be applicable for predicting the risk of HC. Examination of the influence of clinical conditions in a large cohort of HSCT patients has shown that the intensity of the conditioning regimen was the most crucial pretransplant factor affecting HC. The association of MAC and RIC regimens with HC was significantly different in our cohort, as proved by uni- and multivariate analysis and in accordance with other studies. The second risk factor associated with increased HC frequency was male gender (Table 1). Therefore, female HSCT recipients were included in the LCR cohort. The third important factor was associated with prophylaxis of GvHD by a high dose of either posttransplant cyclophosphamide (PtCy) or ATG. This treatment was associated with a high incidence of HC in patients receiving MAC and had a low and insignificant effect on the incidence of HC if it was administered to patients receiving RIC, as shown previously [8,31]. The group of fully matched siblings (MRD) had the lowest incidence of HC due to less intensive GvHD prophylaxis. A lower HC incidence in the group of MRD in comparison with unrelated donors has been reported by many studies [32,33,34,35,36]. Therefore, the male patients in our study receiving any GvHD prophylaxis under the MAC regimen were designated as HCR. All female patients receiving the RIC regimen and patients with prophylaxis lacking ATG or PtCy (MRD) were designated as LCR. The incidence of GvHD had no effect on HC occurrence in our cohort.

Our data (Figure 1B) confirmed the previous findings that BK viruria correlates with the pre-HSCT anti-BKPyV IgG level [21], and that BK viruria >10^7^ copies/mL after HSCT is a very significant risk factor of HC (Table 2) [11,22]. However, pretransplant BKPyV DNAuria is not useful as a pretransplant predictor of HC [37].

The analysis of our data has shown that the risk of HC was higher in younger than in older patients (Table 1). The younger age of adult patients as a risk factor of HC has been reported by several studies [8,33,38,39]. We believe that the predictive power of this association in our study cohort is questionable, given the significantly higher use of MAC than RIC (*p* < 0.0001) in younger patients. Older patients (aged >51) received RIC in 57% of cases, whereas younger patients (aged <51) in 18% of cases only. In the subgroup of patients who received MAC, the median age of patients with HC (43 years) was not statistically different from that of patients without HC (45 years) (*p* = 0.0856). This could be explained by the age distribution in our cohort, as in comparison with other studies the median of age of all patients in this group was much higher.

As a second approach to the prediction of HC, we used the measurement of BKPyV-specific immune response before the start of the conditioning regimen. As for the antibody response, we observed that the HC risk was reduced in patients with the lowest anti-BKPyV IgG levels. This was related to the fact that the group comprised seronegative uninfected patients (OI < 1). Patients with anti-BKPyV IgG levels just above OI value of 1 but below the T1 threshold also had a low incidence of HC. We assumed that the low levels of anti-BKPyV IgG in these individuals could mirror a decreased pretransplant tendency of BKPyV to reactivate, and are possibly due to the waning of antibodies. “R” anti-BKPyV IgG levels between T1 and T2 were associated with the highest risk of HC for the group of recipients at high risk from clinical factors (HCR). HC was less common in patients with the highest anti-BKPyV IgG levels “R_75_” exceeding 75% quartile of positive samples. The predictive value of pretransplant BKPyV-specific IgG was confirmed on the level of virus genotypes. Our results are consistent with a study of humoral responses determined by VLP ELISA (Viracore) in paediatric HSCT patients, where the highest pretransplant anti-BKPyV IgG titers ≥ 1:163,840 were protective against later BK viremia and HC, whereas IgG titers 1:10,240 were associated with a high risk of BK viremia [23]. On the other hand, Lee et al. [22] worked with the same VLP-ELISA and reported that the increase in BKPyV IgG titers correlated with developing BK viruria ≥ 10^7^ copies/mL, and the highest BKPyV IgG titers were not associated with protection. Similarly, Wong et al. [21], using an indirect immunofluorescence assay, observed that adult patients with pre-HSCT anti-BKPyV IgG titer 1:10 had viruria levels up to 10^4^ DNA copies/mL, whereas IgG titers > 1:20 were connected with viruria ≥ 10^8^ DNA copies/mL. In their study, very high levels of IgG had no protective effect.

The data on anti-BKPyV response in donors are rare. Our observation that the immune response of donors did not significantly affect BKPyV reactivation was in line with the results of Wong (2007) [21], who reported that donor’s BKPyV serologic findings did not correlate with BK viruria in the recipient. From our data it seems that the donor/recipient serological mismatch (D−/R+) reported as an important risk factor of cytomegalovirus reactivation in HSCT patients does not have a significant effect in the case of BKPyV [40,41]. To our knowledge, the role of the pretransplant BKPyV-specific T-cell response in the protection from BKPyV reactivation has not been studied so far in HSCT recipients. However, the relevance of posttransplant anti-BKPyV T-cell response for protection was unequivocally proven by the beneficial effect of the adoptive transfer of expanded BKPyV-specific T-cells [17,18] or of donor lymphocyte infusion to patients after HSCT [42]. The importance of BKPyV-specific T-cells for reducing the risk of BKPyV replication was repeatedly demonstrated in kidney transplant recipients [43,44,45].

In our study, the pretransplant anti-BKPyV VP1 CD4^+^ T-cell IFN^+^ response was associated with low BK viruria levels and with a low frequency of HC after HSCT. On the other hand, a weak BKPyV VP1 CD4^+^ T-cell response was associated with high DNAuria after HSCT. The CD4^+^ IFNγ^+^ T-cells of these patients had increased expression of TIGIT, a known marker of exhaustion and senescence that is able to modulate T-cell function during chronic viral infection [46,47,48,49]. It is known that the T-cell response can be altered in haemato-oncological patients as a consequence of leukaemia itself [50] as well as a result of intensive chemotherapy. In such cases, T-cell suppression could contribute to BK virus escape from latency already before transplantation. However, T-cell alteration in haemato-oncological patients is not associated with the BKPyV disease before transplantation, and the unlimited virus replication can start soon after HSCT.

## 5. Conclusions

Our study has shown that the use of a combination of clinical and immunological risk factors can help with the early identification of patients who are at risk of developing the BKPyV disease after HSCT. We found that a specific HC risk can be determined from pretransplant levels of anti-BKPyV-IgG and from the magnitude of the VP1 specific T-cell response. This prediction algorithm could help to accelerate the preparation of BKPyV-specific T-cells for possible adoptive transfer therapy, and improve the follow-up of these patients.

## Figures and Tables

**Figure 2 vaccines-09-01226-f002:**
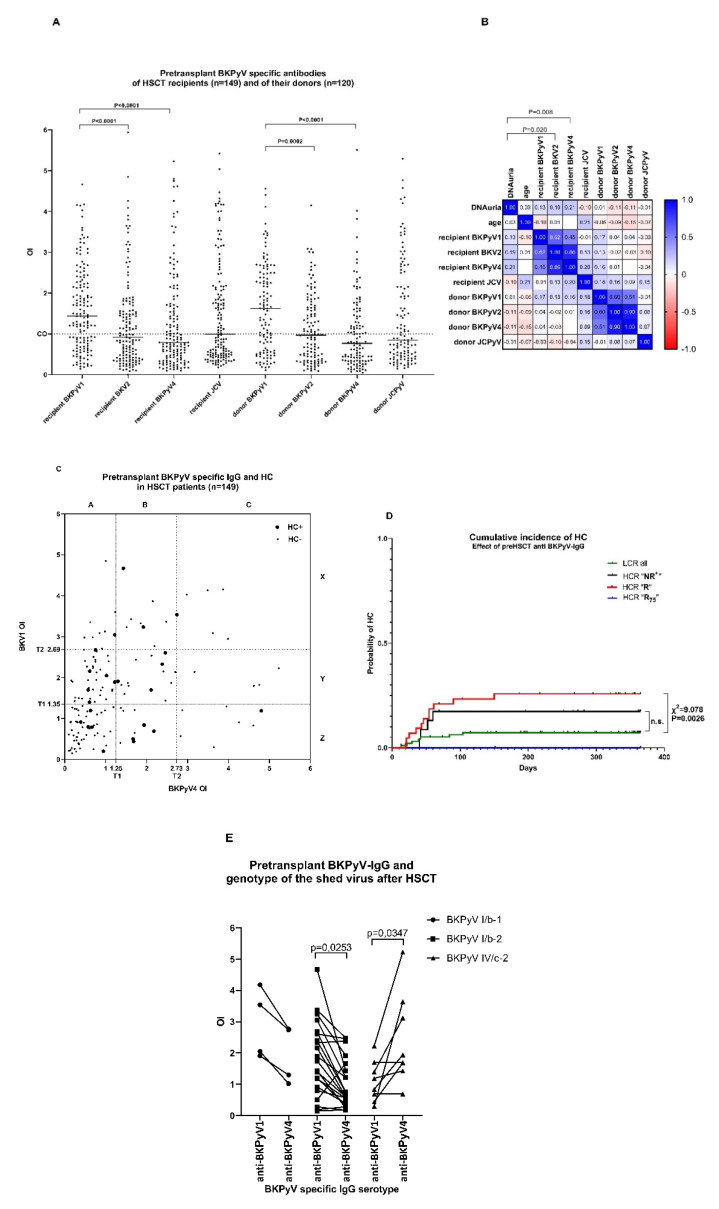
Polyomavirus-specific antibodies of HSCT recipients and donors and their relevance for BKPyV infection in PBIHC study. (**A**) Pretransplant antibodies specific for BKPyV types 1, 2, 4 and JCPyV were measured in serum or plasma by ELISA. Positive samples have OI ≥ 1. The significance of differences between the median of OI was analysed using the paired nonparametric Wilcoxon test. (**B**) Spearman correlation analysis of anti-BKPyV and JCPyV IgG of recipients and donors, maximal DNAuria during the first year after transplantation and their age. The heatmap shows the Spearman correlation matrix and exact Rs values. (**C**). Plot of pretransplant anti-BKPyV 1 and 4 IgG in individual patients of the PBIHC study group (*n* = 149). The large symbols stand for patients with HC. The dotted line starting at T1 represents the guaranteed threshold of positivity for BKPyV 1 and 4 (mean of nonreactive samples of values < OI = 1 + 3 s.d.). The dotted line starting at T2 represents the 75th percentile of all seropositive samples > OI = 1. (**D**) Cumulative incidence of haemorrhagic cystitis (HC) in HSCT recipients stratified according to anti-BKPyV-IgG levels and clinical risk factors (*n* = 149). The HC rate in the HCR group with medium levels of anti-BKPyV IgG was 34.3%. Survival curves were compared using the LogRank (Mantel–Cox) test. (**E**) Pretransplant serotype specific IgG correlate with the genotype of excreted virus after HSCT. The genotype was determined in urine samples of patients with very high BKPyV load >10^9^ (*n* = 33). The significance of the difference between anti-BKPyV1 and anti BKPyV4-IgG levels was determined by the Wilcoxon matched-pairs signed rank test.

**Figure 3 vaccines-09-01226-f003:**
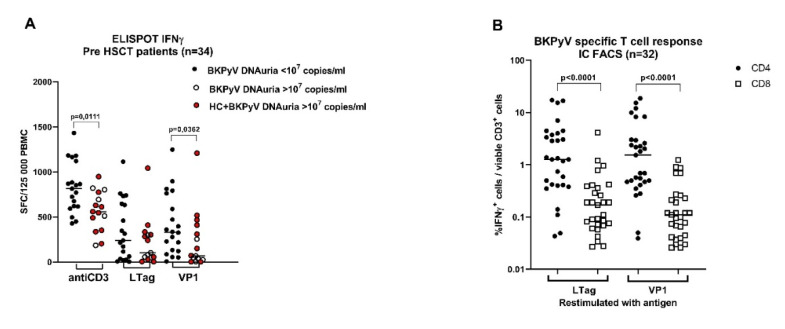
Pretransplant non-specific and BKPyV specific T-cell response in HSCT patients in PBIHC study. (**A**) The response to stimulation with anti-CD3 and BKPyV antigens LTag and VP1 was determined in a subgroup of BKPyV-infected patients by ELISPOT-IFNγ (*n* = 34). Patients were stratified according to the maximal BKPyV DNAuria levels detected during the first year after HSCT. The red circles stand for samples from patients with HC. (**B**) The results of ELISPOT-IFNγ were confirmed by the IC FACS-IFNγ in those patients with sufficient yields of expanded T-cells (*n* = 32). (**C**) The presence of negative checkpoint receptors PD1 and TIGIT was determined by FACS in IFNγ producing T-cells. Analysis was performed using the gating strategy shown in Figure 1. Statistical analysis was performed using the Mann–Whitney test (**A**,**C**) or the Wilcoxon matched-pairs signed rank test (**B**).

**Table 1 vaccines-09-01226-t001:** Retrospective analysis of the risk of HC in a cohort of HSCT recipients transplanted during the period 2014–2020 (*n* = 524) and clinical features of BKPyV reactivation.

				Risk of HC		
				Univariate Associations	Multivariate Associations	
Parameter	HC	Non HC	HC Rate (%)		χ^2^ Test	Logistic Regression
HSCT recipients (*n* = 524)	66	458	12.6			
^a^ Age. median. min-max (year)	46 (20–65)	52 (19–68)		*p* = 0.0057		n.s.
^b^ Gender						
Male	50	273	15.5	*p* = 0.0143 HR = 1.666 95%CI = 1.1084–2.638		OR = 0.4174 95%CI = 0.1868–0.8929
Female	16	185	8.0			
aGvHD						
ref = no aGvHD	39	268	12.7			n.s.
aGvHD grade 1	16	121	11.6	*p* = 0.876		
aGvHD grade 2	10	52	16.1	*p* = 0.5374		
aGvHD grade 3	1	17	5.5	*p* = 0.7093		
^b^ Pretransplant conditioning regimen						
MAC	56	271	17.1	*p* < 0.0001 HR = 0.296 95%CI = 0.155–0.567		OR = 6.099 95%CI = 2.623–15.50
RIC	10	187	5.1			
^b^ Donor HLA match						
Ref = 10/10 MRD	6	86	6.5		*p* = 0.186	n.s.
10/10 MUD with PtCy two doses-50 mg/kg	6	51	9.6	*p* = 0.0537		
10/10 MUD with ATG 20–40 mg/kg	30	152	16.5	*p* = 0.0228 HR = 1.305 95%CI = 1.049–1.51		
Haploidentical	14	103	12.0	*p* = 0.238		
<10/10 MMRD	10	66	13.3	*p* = 0.188		
^b^ Diagnosis						
Ref = non Hodgkin lymphoma	1	25	3.8		*p* = 0.0217 RR = 1.919 95%CI = 1.139–3.881	n.s.
AML	33	200	14.1	*p* = 0.218		
MDS	9	113	7.4	*p* = 0.552		
ALL LBL	9	65	12.2	*p* = 0.446		
CLL	3	14	17.6	*p* = 0.284		
CML	6	10	37.5	*p* = 0.008 HR = 3.000 95%CI = 1.469–5.477		
^c^ Other	5	31	13.8	*p* = 0.386		
Cummulated clinical risk groups HCR vs. LCR						
Significant clinical risk (HCR) Male MAC^+^ 10/10 MUD Male MAC^+^ MMRD	37	121	23.4	*p* < 0.0001 HR = 2.955 95%CI = 1.890 to 4.607		
Low clinical risk (LCR) RIC^+^ All 10/10 MRD All female	29	337	7.9			

^a^ Significance of the difference was determined by Mann–Whitney test. ^b^ Association of HC with risk factors was analysed by the Fisher’s exact test or χ^2^ test. ^c^ Other include: T and NK lymphoma; acute promyelocytic leukaemia; Hodgkin lymphoma; multiple myeloma; Fanconi anaemia; aplastic anaemia; haemophagocytic syndrome; histiocytic sarcoma. Values in bold are statistically significant. Abbreviations: LR—low risk. SR—significant risk. HR—hazard ratio. RIC—reduced intensity conditioning regimen. MAC—myeloablative conditioning regimen. MUD—matched unrelated donor. MMRD—mismatched related donor. AML—acute myeloid leukaemia. MDS—myelodysplastic syndrome. ALL LBL—acute lymphoblastic leukaemia/lymphoblastic lymphoma. nHL—B-non Hodgkin lymphoma. CLL—chronic lymphocytic leukaemia. CML—chronic myeloid leukaemia. AA—aplastic anaemia.

## Data Availability

All data relating to this study are reported in the manuscript and figures.

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
