# Peer review of "Pretransplant BK Virus-Specific T-Cell-Mediated Immunity and Serotype Specific Antibodies May Have Utility in Identifying Patients at Risk of BK Virus-Associated Haemorrhagic Cystitis after Allogeneic HSCT"

_vaccines, 2021, doi:10.3390/vaccines9111226_

Round 1
Reviewer 1 Report
Stastna-Markova et al. evaluate the risk of haemorrhagic cystitis in a cohort of 149 HSCT recipients in relation to BKPyV specific immunity.
This study represents a lot of work. However, the presentation of the results is not always very clear and the conclusions are a bit forced in relation to the results of the study.
Normally, hemorrhagic cystitis is an immune reconstitution pathology with a reaction of the donor's immune system to the recipient's viral strain which during the induction phase has actively replicated. The T cell response was only assessed in 34 recipients (Figure 3A) and not in donors?
Is it not possible to have this data?
-From lines 198 to 202 is mentioned "fig 1A" to what does this correspond?
- from lines 297 to 299, the announced scores are not found in Fig 2B
- The serological tests were only performed at 1/100 dilution. Wouldn't it have been interesting to do additional dilutions in order to have titers rather than index comparison
- Lines 354-356 the 13% score is not found in Figure 2D (blue)?
- Figure 2E is not described in the text and difficult to understand.
Author Response
Author´s Reply to the Reviewer 1
- …The T cell response was only assessed in 34 recipients (Figure 3A) and not in donors?
In our study the majority of grafts was obtained through foreign BM registers. The cells from this source must not be used for research purposes and were not available for our study. Therefore, T cell responses were measured in patients only. The detection of pretransplant T cell responses of recipients started in the later stage of study when only 34 seropositive recipients were available. The data from a larger group of recipients would allow to test how T cell responses affect not only virus loads, but also HC. A comment on etiology of HC in connection with immune reconstitution syndrome was inserted (line 344-350).
- . -From lines 198 to 202 is mentioned Fig. 1A
We apologize for errors that contributed to difficulties with manuscript reading. Fig. 1A has been changed to Fig. 2C (lines 203, 205, 209).
- …..-from lines 297-299, the announced scores are not found in Fig. 2B
The medians of IgG BKPyV 1,4 for young and old age groups were added in the text (lines 304-306). The result of correlation analysis of age is in Fig. 2B.
- The serological tests were performed only at 1/100 dilution. Wouldn´t it have been interesting to do addition dilutions in order to have titers rather than index comparison.
Yes we agree. Our ELISA test is semi-quantitative only. We determined by testing samples in dilutions 1/100 and 1/500, that ELISA performance in samples of OI range up to 2.0 is linear. This is important for discrimination between “NR+” and “R” levels of IgG.
- -Lines 354-356 the 13% score is not found in Figure 2D (blue)?
The 13% score in Table 2 concerns “R75” group from all 149 recipients. Figure 2D depicts incidence in groups stratified according combination of clinical and serological risk factors. The blue line represents HCR “R75” in Table 2 with 0%.
- – Figure 2E is not described in the text and difficult to understand.
Again, an error in numbering of Figures for which we apologize. Former Fig. 2C is now changed to Fig. 2E (line 358). One sentence summarizing the result was inserted (lines 361-363).
- Our manuscript has been checked by a native English-speaking colleague.
Reviewer 2 Report
I really enjoyed reading this paper that is very well written, and based on a very well designed study project. I found the conclusions very interesting and I think that the manuscript deserves to be published. I have a main concern regarding the measurement of JCPyV IgG and its significant, so I am asking the authors to better explain the significance.
Minor point: JCV should be JCPyV all over the text
Author Response
Author´s Reply to the Reviewer 2
- …..concern regarding the measurement of JCPyV IgG and its significancy…..
JCPyV and BKPyV are related viruses with significant homology between capsid proteins. Subsequences of protein similarity include cross-reactivity of immune sera against VLPs detected by ELISA and cross-recognition of VP1 derived peptide epitopes by T cells. One of our aims was to determine whether donor JCPyV specific immunity contributes to protection against reactivation of BKPyV infection and HC. Our explanation was inserted in the text (lines 350-353).
- JCV should be JCPyV all over the text
Virus name was emended (lines 154 and 169)
- Our manuscript has been checked by a native English-speaking colleague.
Round 2
Reviewer 1 Report
na